# Time-dependent changes in stepping performance and velocity following partial dopaminergic lesions in the SNc of male and female rats

Diego Lievano Parra[1,2]*, Juan David Garavito Coronado[1‡], Greg Jensen[2‡],
Valeria Gonzalez Diaz[2], Fernando Cardenas Parra[1]

1 Behavioral Neuroscience Laboratory, Psychology Department, University of Los Andes, Bogota,
Colombia, 2 Psychology Department, Reed College, Portland, Oregon, United Sates of America

☯ These authors contributed equally to this work.
‡ JDGC and GJ also contributed equally to this work.
* dj.lievano@uniandes.edu.co

## Abstract

Parkinson's disease (PD) is a neurodegenerative disorder in which gait disturbances
are a major source of disability, yet their early manifestations remain difficult to char-
acterize because prodromal PD is challenging to identify in humans. Rodent models
may help address this gap by approximating partial dopaminergic loss and enabling
investigation of motor changes that precede overt symptoms. Here, we exam-
ined stepping performance and velocity in Wistar rats following unilateral low-dose
6-hydroxydopamine (6-OHDA) lesions in the substantia nigra pars compacta (SNc).
Animals were assessed weekly for six weeks in the horizontal ladder, where velocity
and footfall errors were quantified using markerless pose estimation with DeepLab-
Cut (DLC). Lesioned animals showed greater tyrosine hydroxylase (TH) asymmetry
than controls, accompanied by persistent but gradually attenuating impairments in
stepping accuracy. Footfall errors peaked around week three and remained elevated
thereafter, whereas velocity showed more heterogeneous changes across time.
Although sex effects were not uniformly robust, control females showed an increase
in velocity across weeks, whereas lesioned females exhibited a less pronounced and
more variable pattern. These findings indicate that partial SNc lesions can reveal
subtle and time-dependent alterations in locomotor skills relevant to the study of
early dopaminergic dysfunction. Combined with DLC-based tracking, this framework
provides a practical approach for detecting early behavioral changes and refining the
study of preclinical motor features of PD.

doi.org/10.1371/journal.pone.0337381

University Carbondale, UNITED STATES MINOR
OUTLYING ISLANDS

**Peer Review History:** PLOS recognizes the
benefits of transparency in the peer review
process; therefore, we enable the publication
of all of the content of peer review and
author responses alongside final, published
articles. The editorial history of this article is
available here: https://doi.org/10.1371/journal.
pone.0337381

**Data availability statement:** All relevant data are included within the manuscript and its Supporting Information files. The full repository (including videos, databases, and DLC model code) is publicly available from Zenodo at 10.5281/zenodo.17518250.

**Funding:** The author(s) received no specific funding for this work.

**Competing interests:** The authors have declared that no competing interests exist.

## Introduction

Parkinson's disease (PD) is a neurodegenerative disorder that affects approximately 2% of individuals over the age of 60 and is clinically characterized by resting tremors, rigidity, bradykinesia, and gait abnormalities [1,2]. While these motor symptoms are well described in advanced stages [3], the impact on motor behavior during early stages remains poorly understood. Identifying the onset of neurodegeneration is particularly difficult because pathophysiological changes may precede quantifiable motor abnormalities. Because progressive dopamine depletion is a defining pathological feature of PD, rodent models that replicate partial loss of dopaminergic neurons can offer unique opportunities to investigate motor alterations that precede overt clinical diagnosis.

Among these models, the 6-hydroxydopamine (6-OHDA) model has been widely used to mimic PD-like motor deficits in rodents, showing a clear dose-dependent relationship between dopamine depletion and motor impairment [4]. Traditionally, high-dose of 6-OHDA have been used to produce extensive lesions, reliably reproducing the motor deficits that resemble advanced PD [5,6]. Nevertheless, these approaches provide limited insight into the subtle and dynamic behavioral changes that accompany early stages of dopamine depletion.

To address this limitation, low-dose 6-OHDA protocols can induce moderate lesions that mimic mild, variable, and adaptive motor deficits, offering a closer approximation to early stages of PD [7]. Furthermore, targeting the substantia nigra pars compacta (SNc) is advantageous, as it captures the earliest site of dopaminergic vulnerability in PD [8]. Nevertheless, behavioral outcomes of moderate lesions can be variable, due to factors such as lesion site, dosage, assessment window, and subject characteristics [9]. For instance, moderate dopaminergic depletion in the medial forebrain bundle (MFB) and dorsal striatum (DS) often leads to measurable gait impairments [10], whereas similar depletion in the SNc may produce subtler effects [11]. This variability underscores the importance of refining SNc-targeted lesions, as they offer a promising framework for modeling motor patterns that resemble early PD.

In addition, most studies have focused on male rodents, leaving potential sex-related variation in early locomotor responses underexplored, despite evidence that factors beyond dopaminergic cell loss can influence plasticity, compensation, and motor outcomes [12]. Consequently, experimental designs that often include only males may provide an incomplete view of how early behavioral changes emerge after partial dopaminergic injury. Addressing this gap requires wider experimental approaches to detect subtle locomotor changes over time while allowing sex to be considered as a biologically relevant source of variability.

Although traditional gait analysis in PD has often relied on either discrete markers or sensors to facilitate motion tracking, or on manual review of video recordings [13,14], both approaches have important limitations. They may introduce observer bias and may not capture subtle or transient locomotor changes consistently across repeated assessments. More recently, automated markerless pose estimation methods such as DeepLabCut (DLC) have been increasingly adopted in the study

of locomotor activity, showing high sensitivity to subtle gait changes [15–17]. Importantly, the value of these tools is not limited to highly detailed kinematic reconstruction. Even under relatively simple recording conditions, they can improve the standardization and objectivity of behavioral measurements and support the quantification of locomotor outcomes such as footfall errors and velocity. This makes them particularly useful for studies aiming to detect early motor alterations in animal models under accessible experimental conditions, especially in settings where specialized motion-capture systems are not available. This study aimed to characterize early alterations in footfall errors and velocity following partial dopaminergic lesions induced by low-dose 6-OHDA in the SNc of male and female Wistar rats. Locomotor behavior was tracked over six weeks to examine the temporal evolution of footfall errors and velocity in relation to TH-defined lesion status. To achieve this, we used markerless pose estimation with DeepLabCut (DLC) to quantify stepping accuracy and locomotor progression under standardized recording conditions. This approach was intended to improve the detection of subtle locomotor changes while providing a practical framework for studying early motor dysfunction in rodent models relevant to prodromal Parkinson's disease.

## Materials and methods

### Animals

Subjects were 101 Wistar rats (*Rattus norvegicus*), 57 males and 44 females, were bred at the Neuroscience and Behavior Laboratory at Universidad de Los Andes in Bogotá, Colombia. Subject ages were between postnatal days (PND) 60 and 80 at the start of the experiment. Subjects were randomly paired in standard laboratory cages measuring $16.5 \times 50 \times 35$ cm. Water and food were provided ad libitum during the experiment. Environmental conditions were consistent throughout the study: a 12−12 hour light/dark cycle (lights on at 14:00), a temperature of $22 \pm 2$ °C, and a relative humidity of $57 \pm 10\%$. Experiments were conducted during the dark portion of the cycle. The experimental protocol was approved by and followed the guidelines established by the Institutional Committee for the Care and Use of Laboratory Animals (CICUAL) at Universidad de Los Andes (Approval number CICUAL_20_012, issue date May 18, 2020), and adhered to the national ethical regulations for animal research in Colombia (Law 84 of 1989 and Resolution 8430 of 1993, Ministry of Health).

### Surgery and 6-OHDA lesion

Subjects were randomly assigned to receive either 6-hydroxydopamine (6-OHDA) or a vehicle solution. Rats were anesthetized with Isoflurane, craniotomies were created, and a 27-gauge needle was inserted unilaterally in the substantia nigra compacta (SNc) following the stereotaxic coordinates by Paxinos and Watson (2007): AP −5.25 mm, ML 2.2 mm, and DV 7.8 mm from Bregma. 6-OHDA solution (Sigma-Aldrich) was prepared at a concentration of 10 µg/µl in 0.9% saline contained ascorbic acid 0.02% w/v ($\approx 0.2$ mg/mL). A total volume of 0.25 µl (2.5 µg) was infused into the SNc via a 10 µl Hamilton™ syringe linked to a NE-1000 microinjection system (New Era Pump Systems Inc.) at a rate of 0.125 µl/min (infusion duration $\approx 2$ min). After infusion, the needle was left in place for 5 additional minutes to allow diffusion. Control subjects received an equal volume of 0.9% Vehicle. Postoperative management involved 3–5 days of Meloxicam® administered subcutaneously.

### Experimental design

One week prior to surgery, all subjects were handled daily for 10 min to facilitate habituation. Baseline motor performance was assessed on day six using the horizontal ladder, and surgeries were performed the following day. After surgery, animals were randomly assigned to one of three cohorts defined by the post-lesion interval at which histological assessment was performed: two weeks (t1), four weeks (t2), or six weeks (t3) (Fig 1A). This design enabled evaluation of tyrosine hydroxylase (TH) immunoreactivity at multiple post-lesion time points. The distribution of animals across sex, treatment, and cohort is summarized in Table 1.

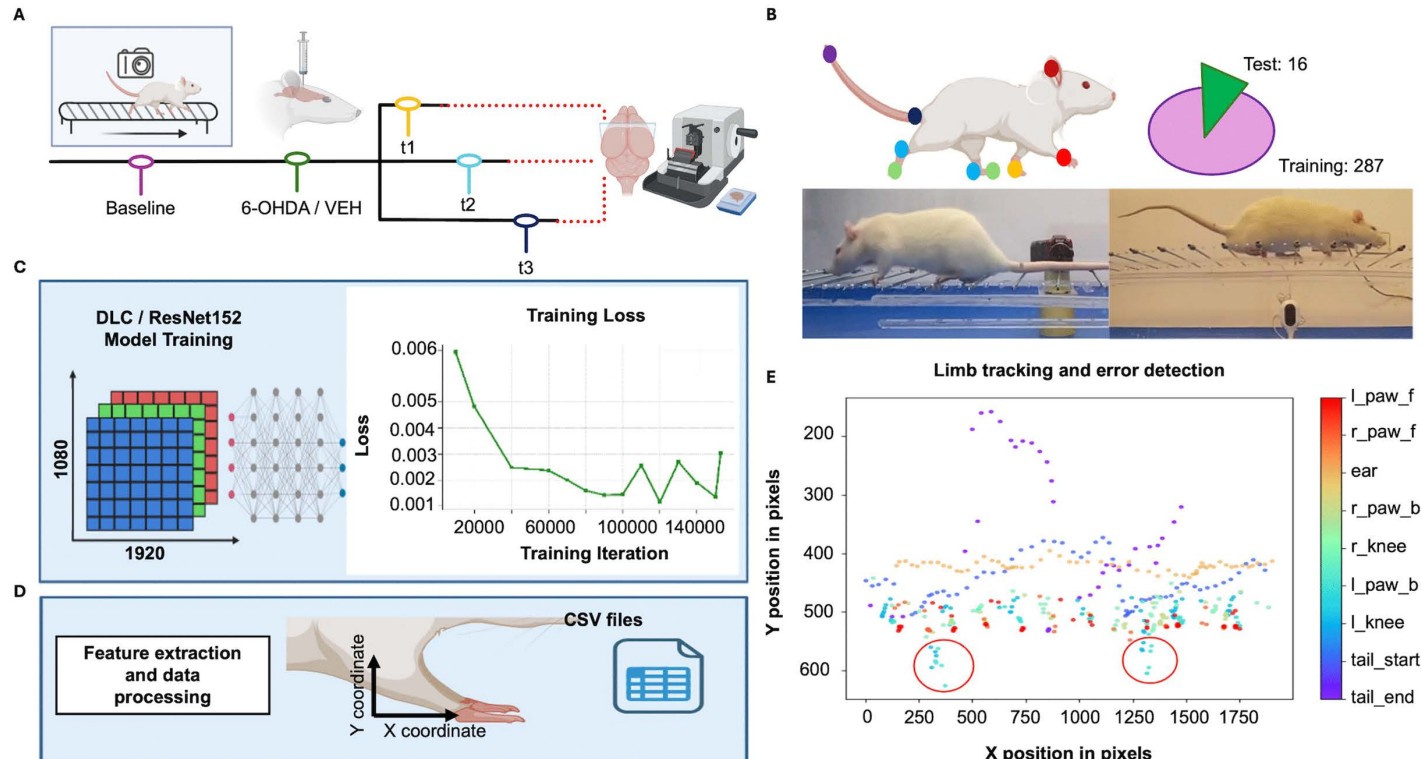

**Fig 1. Schematic overview of the workflow applied to horizontal ladder walking recordings for the extraction of footfall errors and velocity measurements. A**. *Experimental design*: Subjects unilaterally received either vehicle or 6-OHDA in the SNc and were subsequently assigned to one of three cohorts for locomotor evaluation over 2, 4, or 6 weeks. After each cohort completed its respective behavioral assessment, tissue was collected, stored, and processed. **B**. *Data acquisition*: Videos were recorded using a Nikon camera positioned laterally to the ladder, capturing the right side of the rat during each trial. **C**. *AI model design*: A total of 303 frames were manually labeled with 9 anatomical keypoints. Frames were selected from multiple videos to capture a representative range of postures, locomotor states, and gait phases, including running, standing, missteps, and hopping. Landmark annotation consistency was independently reviewed to ensure accurate and reproducible placement across frames. Annotations were assigned only when the corresponding body part was fully visible; frames with occluded or out-of-view body parts were excluded. The dataset was split into 287 frames for training and 16 for testing. **D**. A pose estimation model was trained using DLC with a ResNet-152 backbone. The framework leverages transfer learning from networks pre-trained on large image datasets, which reduces the number of annotated frames required for accurate pose estimation. The resulting X and Y coordinates for each body part were exported as.csv files. **E**. *Database creation and analysis*: Two behavioral metrics were extracted from the dataset: (1) the number of footfall errors, defined as displacements exceeding a paw-specific threshold and occurring in frames with a likelihood score > 0.75, and (2) the velocity in the X-direction, calculated using the displacement of the ear across frames, initially extracted in pixels per second (pix/second) and subsequently transformed and reported in centimeters per second (cm/s).

**Table 1. Sample distribution across cohorts by sex and treatment.**

| Cohort | Male Control | Male Lesion | Female Control | Female Lesion | Total |
|--------|--------------|-------------|----------------|---------------|-------|
| **t1** | 9 | 11 | 6 | 7 | 33 |
| **t2** | 7 | 9 | 6 | 7 | 29 |
| **t3** | 7 | 14 | 9 | 9 | 39 |
| **Total** | 23 | 34 | 21 | 23 | 101 |

Table 1 Cohorts correspond to the terminal time point at which animals were perfused for histological analysis, at two, four and six weeks, respectively.

Behavioral testing was conducted weekly following surgery. Animals in each cohort were evaluated longitudinally until their designated endpoint, after which they were perfused for immunohistochemical analysis. Because cohorts were collected at different post-lesion intervals, the number of animals contributing behavioral observations decreased across weeks according to the tissue collection schedule. For behavioral analyses, assessments at weeks 1–2 included animals from all three cohorts (t1, t2, t3), at weeks 3–4 included animals from the t2 and t3 cohorts, and at weeks 5–6 included only animals from the t3 cohort.

## Gait measurements in the horizontal ladder

The structure of the horizontal ladder consisted of a transparent acrylic corridor measuring $100 \times 15 \times 62$ cm (L×W×H), with steel rungs spaced 4 cm apart (Fig 1A). A home cage was positioned at one end of the ladder and connected via a short ramp, allowing animals to enter a familiar environment after completing the ladder crossing. At the beginning of each trial, rats were placed at the opposite end of the ladder and allowed to cross the entire length toward the home cage. Trials ended when the animal reached the cage. No time limit was imposed; if an animal paused while the crossing, gentle tactile stimulation of the back was used to encourage movement. Each subject completed up to three trials per weekly session, separated by 2-min intervals. For analysis, the trial with the fewest stops was selected for analysis, and when a subject completed the first uninterrupted ladder crossing, that trial was used. This approach was intended to capture performance during continuous locomotion while reducing variability introduced by exploratory pauses unrelated to stepping execution.

Given that videos were recorded from a single lateral view, analysis was limited to measures that could be obtained robustly and interpreted reliably under those acquisition conditions. For this reason, gait performance was quantified using footfall errors and velocity, which indexed stepping accuracy and locomotor progression, respectively. Footfall errors were defined as instances in which a paw slipped off a rung during ladder crossing. Velocity was defined as the distance covered across the apparatus divided by trial duration. Trial onset corresponded to the first frame in which the animal was stably positioned on the ladder with their four paws and initiated forward movement, and trial end was defined as the frame in which the ear landmark reached the end of the apparatus. Duration was calculated from the total number of frames at a recording rate of 30 fps. Distance was estimated from horizontal displacement and converted from pixels to centimeters using the known 4-cm spacing between adjacent rungs, allowing velocity to be expressed in cm/s. All trials were video recorded individually and analyzed offline.

## Data acquisition

Video recordings were obtained using either an iPhone X and a Nikon Coolpix B500 camera positioned laterally and orthogonal to the horizontal ladder at a fixed distance to maintain a consistent field of view. Early recordings captured with a fisheye lens were excluded from the training dataset to prevent distortion-related errors. High-definition videos ($1920 \times 1080$ pixels) were recorded at 30 fps, providing a right-side lateral view of locomotor activity (Fi 1D). A total of 303 frames (287 for training, 16 for testing) were manually selected from sixteen videos to ensure representative coverage of postures and paw movements, including running, standing, missteps, and hopping. Frame selection was targeted to sample multiple limb positions across the locomotor cycle, including paw contact, limb lift, and forward swing, although frames were not formally stratified into predefined gait phase categories. Nine anatomical landmarks were annotated using the DeepLabCut (DLC) graphical user interface (GUI): left and right hind paw joints (ankle, knee), both forepaws, tail base and tip, and right ear (Fig 2A). All frames preserved the original resolution and orientation; no cropping or downsampling was applied.

## Immunohistochemistry

At the conclusion of each cohort motor evaluation interval, animals were euthanized with an overdose of sodium pentobarbital (85 mg/kg, i.p.) and transcardially perfused with 100–150 ml of 0.1 M phosphate-buffered saline (PBS), followed by 100–150 ml of 4% paraformaldehyde (PFA). Brains were extracted, post-fixed in PFA at 4°C, and coronally sectioned at 20 µm using a Compresstome VF-300-0Z microtome. Sections were stored in 0.1 M PBS at 4°C until further processing.

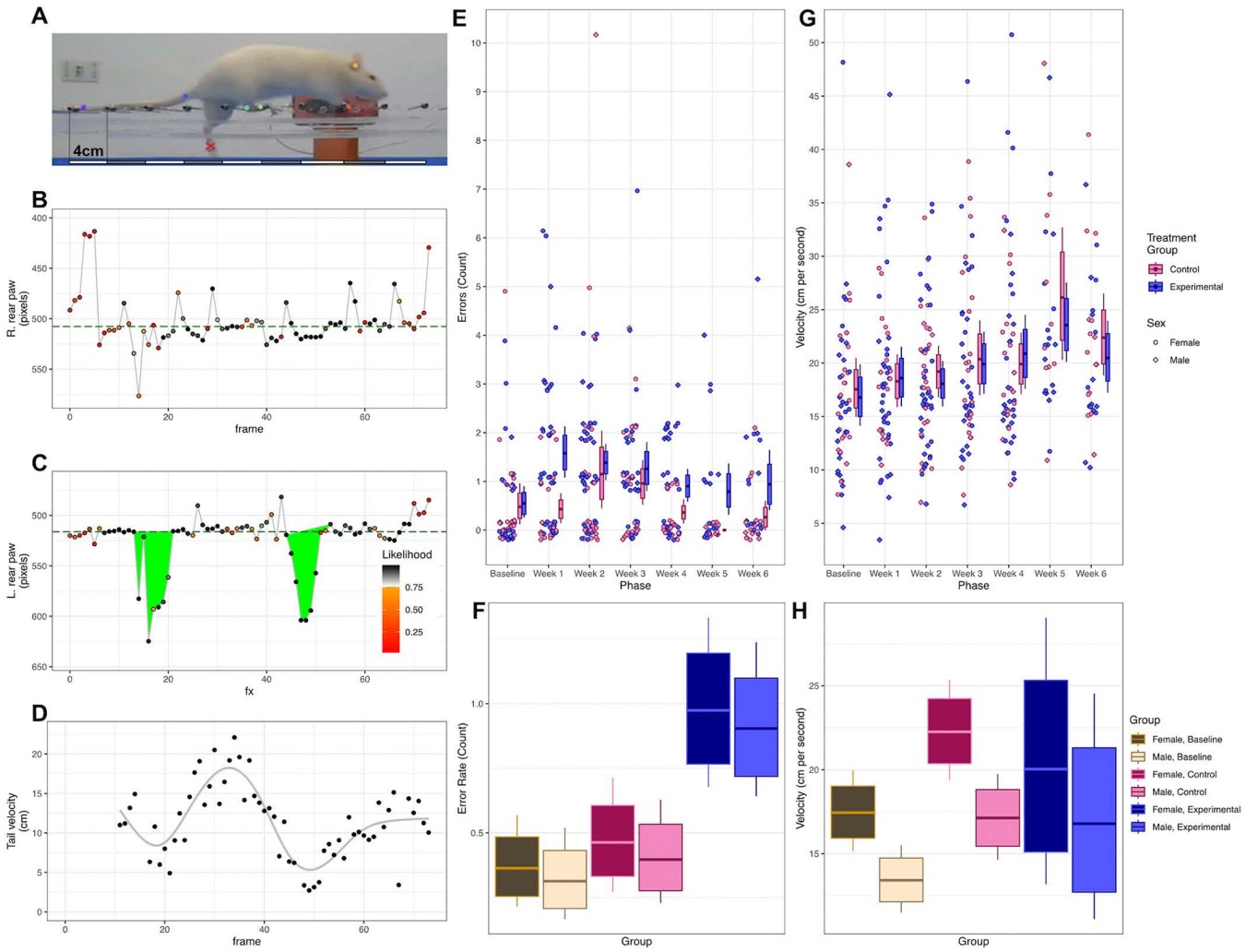

**Fig 2. Analysis of step errors and velocity. A.** Estimated anatomical tags overlaid on a representative video frame, according to DeepLabCut. The red X is a manually classified footfall error. **B.** Estimated horizontal position of the back right paw in each frame of a representative video. Points are color-coded according to each estimate's likelihood. Only grayscale points (LH > .75) were "confirmed" and used for classification. No footfalls with this paw were classified as errors. **C.** As in panel B, for the back left paw. Two footfall errors were detected, filled in green. Errors began with the first confirmed position before threshold, and ended with the first confirmed position above baseline. **D.** Estimated interpolated velocity for confirmed foot positions. Also shown is a representative curve (generalized additive model, solid gray line), and overall mean velocity (total distance/total time, dashed green line). **E.** Footfall errors (circles = females, diamonds = males) per week by the control (pink) and experimental (blue) groups. Bootstrapped means are reported (bars = 80% confidence intervals, whiskers = 95% confidence intervals). Points represent discrete counts, but were randomly jittered on both axes to assist visual inspection. **F.** Summary of estimated error rates for an average subject in each condition (tan = baseline, pink = control, blue = experimental), for both sexes (darker colors = female, lighter colors = male). Posterior estimates were computed using Equations 1-4 (see S2 for more details), limited to only fixed effects (all $\beta$ terms) and the population mean across subjects ($\eta$). Estimates for the control and experimental conditions here averaged performance across all six weeks (bars = 80% posterior credible intervals, whiskers = 95% credible intervals). **G.** As in panel E, but plotting overall velocities (cm/second). Points were jittered on the horizontal axis to assist visual inspection. **H.** As in panel F, but plotting overall velocities (cm/second).

Coronal sections containing the substantia nigra pars compacta (SNc) were located using the Paxinos and Watson (2007) rat brain atlas (AP, −5.25 mm from bregma; ML, 2.2 mm; DV, 7.8 mm). From a subset of animals in each condition (2–3 subjects per group), nine sections spanning the rostrocaudal extent of the SNc were collected. From these, three anatomically matched sections at comparable anteroposterior levels across animals were processed for tyrosine

hydroxylase (TH) immunohistochemistry to allow standardized hemispheric comparisons. When the initial staining run did not yield a sufficient number of usable sections in all cases, the procedure was repeated using the remaining samples to obtain adequate material for asymmetry index quantification.

For TH immunohistochemistry, free-floating sections were rinsed twice in 0.1 M PBS (10 min each). Endogenous peroxidase activity was quenched with hydrogen peroxide blocking solution for 10 min at 15–18°C, followed by three PBS washes. Sections were then incubated in protein blocking solution for 10 min and rinsed twice with PBS before overnight incubation at 4°C with anti-TH primary antibody (Abcam, ab6211; 1:1000). On the following day, sections were rinsed four times in PBS and incubated with secondary antibody solution for 20–30 min using a rabbit-specific HRP/DAB detection kit (Abcam, ab64261). After four additional PBS washes, immunoreactivity was visualized with diaminobenzidine (DAB; 1–10 min), followed by three final washes in PBS. Sections were mounted on glass slides and coverslipped with canada balsam.

## TH quantification and asymmetry index

Digital photomicrographs were acquired separately from the ipsilateral and contralateral SNc using a Canon EOS Rebel T3i camera mounted on a Meiji MT5000 microscope with a 10× objective under identical illumination and exposure settings. SNc regions of interest were manually delineated in ImageJ (NIH) using atlas-defined anatomical boundaries and consistent cytoarchitectural landmarks across hemispheres and animals. Within each ROI, TH-positive neuronal somas were identified and counted independently by two evaluators under single-blind conditions using the Cell Counter tool in ImageJ. Only soma-like TH-positive profiles were included; fibers and diffuse staining were excluded. Because the number of usable sections varied across animals, counts were first averaged across sections within each animal and hemisphere, thereby preserving the animal as the unit of analysis. Inter-rater agreement was assessed at the animal-by-hemisphere level using two-way absolute-agreement intraclass correlation coefficients (ICC) and Bland–Altman analysis; details are provided in S1 File (Assessment of inter-rater agreement). Final hemisphere-specific values were defined as the mean of the two raters.

Hemispheric asymmetry was quantified on a within-subject basis using an asymmetry index (AI), defined as $AI = \frac{I}{C}$, where $I$ denotes the injected hemisphere and $C$ the contralateral side. An AI of 1 indicates equal counts across hemispheres, whereas values <1 indicate reduced TH-positive counts in the injected hemisphere. Lower values indicate greater ipsilateral depletion. The index was derived from direct counts of TH-positive somatic profiles rather than optical density measurements. Stereological methods were not used because the goal was to quantify relative hemispheric depletion within each animal rather than absolute neuronal number.

For inferential analysis, the primary outcome was the log-transformed asymmetry index, $log(AI) = log\left(\frac{I}{C}\right)$, for which hemispheric symmetry corresponds to 0 and increasingly negative values indicate greater reduction in the injected hemisphere. The outcome was analyzed using a linear model including treatment, time, and their interaction:

$$log(I/C)_i = \beta_0 + \beta_1(treatment)_i + \beta_2(time)_i + \beta_3(treatment \times time)_i + \varepsilon_i,$$

where treatment was coded as Vehicle or 6-OHDA, time corresponded to cohorts t1, t2, or t3, and $\varepsilon_i$ denotes the residual error term. Because sample sizes were small and unbalanced across sex within treatment-by-time cells, sex was not included in the primary model. Sensitivity analyses were performed using the raw interhemispheric difference, $I - C$, and the relative preservation of the injected hemisphere, expressed as $AI \times 100 = \left(\frac{I}{C}\right) \times 100$. Estimated marginal means were used to compare control and lesion groups within each time point, with Holm correction for multiple comparisons.

## Statistical analyses

**Video analyses and DLC performance.** Pose estimation was performed using DLC versions 2.3.10 and 2.3.11 with a ResNet-152 backbone pre-trained on ImageNet (Fig 1C). This architecture was selected because transfer learning with a deeper backbone provided stable landmark detection under conditions of motion blur and rapid limb movement. The

network was trained for 160,000 iterations using stochastic gradient descent. The effective batch size was set at 4 due to GPU memory constraints. Training was conducted on Google Colab with NVIDIA T4 and A100 GPUs (16 GB and 40 GB VRAM, respectively), running Python 3.11 and CUDA 11.8. Model accuracy was quantified as mean absolute error (MAE) in pixels between predicted and manually annotated landmark position for both training and test sets. After training and validation, the DLC model was applied to the remaining video dataset. For each frame, X–Y coordinates were extracted for all annotated landmarks (Fig 1D). To minimize false positives, landmarks with a likelihood score below 0.75 were excluded from their corresponding frames prior to evaluation. MAE was computed both before and after likelihood filtering, the filtered MAE reflected higher confidence detections, ensuring that localization error estimates were not biased by low-confidence predictions.

Footfall errors were independently calculated for each paw by determining a vertical baseline from a trimmed mean of Y-coordinates. A candidate error was defined as beginning with a downward excursion beyond an adaptive threshold, calculated as the paw's baseline plus a fixed offset (35 pixels in our implementation). This adaptive approach ensured that thresholds were referenced to each paw's height, accounting for differences in limb perspectives across recordings [16,18]. This error footfall was then considered to continue until the first foot position confirmed (LH > 0.75) to be above baseline. Consolidating all such frames into a single error epoch prevented double-counting of transient fluctuations during the same footfall (Fig 2B and C). For each footfall error, maximum drop distance and epoch duration (frames) were recorded.

Horizontal velocity was estimated using the X-coordinate of the ear landmark, which showed the lowest incidence of occlusion (Fig 2D). Average velocity (pixels/s) was calculated as displacement over time for the first half, second half, and total duration of each trial. Time was reconstructed from frame counts assuming a constant acquisition rate of 30 frames per second (fps). To convert velocity to centimeters per second (cm/s), a pixel-to-centimeter calibration factor was obtained for each video. The pixel distance between adjacent bars, corresponding to a known physical distance of 4 cm, was measured using *Adobe Illustrator* (Adobe Inc., San Jose, CA, USA). A single frame per video was extracted, and the horizontal distance (in pixels) between the centers of adjacent bars was measured six times: twice on the left side, twice in the center, and twice on the right side of the apparatus. The mean of these measurements was used to account for perspective-related variation and reduce measurement error. The resulting conversion factor (cm/px) was then applied to the velocity values by multiplying px/s by cm/px, yielding velocity in cm/s.

## Behavioral analyses

Three behavioral measures were assessed (total footfall errors, proportion of left-side footfall errors, and velocity) using multi-level Bayesian regression models, implemented using the Stan programming language [19]. All models used the same overall structure, characterizing performance for a given subject s during a particular week was a sum of fixed and random effects, given by $\mu_{s,w}$:

$$\mu_{s,w} \;=\; \gamma_s \;+\; X_s \cdot \beta_x \;+\; T_s \cdot X_s \cdot \beta_{tx} \;+\; W_s \cdot \beta_w \;+\; T_s \cdot (W_s \cdot \beta_t) \tag{1}$$

Here, $\gamma_s$ denotes a random intercept specific to subject *s* that was constant across the experiment. $X_s$ denotes a centered dummy variable for the subject's sex (female = −0.5, male = 0.5) and $T_s$ denotes a dummy variable for treatment (control = 0, experimental = 1). $\beta_x$ thus denotes a fixed effect of sex and $\beta_{tx}$ denotes a sex-by-treatment interaction term. $W_s$ is a vector of six dummy values specifying which week was being modeled during the experiment; only one value in $W_s$ was set to 1 at a time, with the rest set to zero. To measure baseline performance, all six values in $W_s$ were set to zero. Finally, $\beta_w$ denotes a vector of six fixed effects for each week of the experiment (constituting that week's difference from baseline performance), and $\beta_t$ denotes six fixed effects of treatment during each week (constituting the difference between experimental and control conditions during that week). Thus, in total, performance was modeled using one random parameter per subject, as well as 14 fixed effects.

Using the structure described above, total errors for a given subject during a given week were modeled as a Poisson process governed by a rate $\theta_{s,w}$, which was related to $\mu_{s,w}$ via a log link, such that $\theta_{s,w} = \exp(\mu_{s,w})$. Full details pertaining to the model structure and priors are provided in the S2 File (Bayesian Modeling).

The decision to model weeks as a nominal variable, making no assumptions about how similar adjacent weeks might be, was motivated by a desire to allow for non-linear patterns to emerge (such as an initial disruption of performance, followed by improvement with increased experience). This allowed the model maximal flexibility to reveal patterns in the data without making unnecessary assumptions.

## Results

### DLC model performance and application to behavioral dataset

The ResNet-152 network trained on 303 curated frames achieved a test set MAE of 8.41 px without thresholding, which improved to 5.42 px with a p-cutoff $\geq 0.75$. Training MAE was 3.80 px without the cutoff and 3.32 px with it (Fig 2B-C). These results confirmed high pose estimation accuracy suitable for downstream gait analysis. The trained model was then applied to the complete set of experimental videos. Error analysis revealed limb-specific deviations consistent with missteps, meeting all three detection criteria (baseline deviation, high confidence, and single-event consolidation). Average velocity was calculated for the first half, second half, and entire trial, allowing temporal segmentation of locomotor performance. Minor variability in the pixel-to-centimeter calibration factor was observed within some frames, likely reflecting small differences in camera positioning. However, measurements were largely consistent across weeks and spatial regions of the apparatus, indicating stable pixel scaling and supporting the reliability of the conversion to cm/s. Taken together, these outputs provided the basis for statistical analyses evaluating differences between lesioned and control groups throughout the study.

### Behavioral results

There was considerable variation both between subjects and across sessions, as illustrated in Fig 2E, which plots the errors made by each subject during each session. The modal outcome for any given phase of the experiment was zero errors, with considerable positive skew. Bootstrapped means per condition suggest that 0.5 to 1.0 errors per session are a typical rate. The corresponding velocities also showed some positive skew and suggested a gradual acceleration across weeks, as illustrated in Fig 2G.

In order to estimate the effects in a manner that controls for covariates, parameters were fit for the models described in Equations 1–4. Overall performance for each of the conditions (pre-treatment, control, and experimental), averaged across all weeks, given an average subject of each sex, is summarized in Fig 2F. At baseline, an average subject was expected to make 0.331 errors [$CI_{95\%}$: 0.208, 0.491], compared to 0.425 errors [$CI_{95\%}$: 0.277, 0.605] in the control condition and 0.934 errors [$CI_{95\%}$: 0.715, 1.184] in the experimental condition. These results indicate that the experimental treatment resulted in about 0.508 additional errors [$CI_{95\%}$: 0.255, 0.776] than did the control treatment. However, a reliable difference in mean errors did not arise as a function of sex. Fig 2H depicts a similar summary for velocity. This shows that, at baseline, an average subject was expected to traverse the apparatus at 15.32 cm per second [$CI_{95\%}$: 13.32, 17.73], compared to 19.69 cm per second [$CI_{95\%}$: 17.01, 22.55] in the control condition and 18.41 cm per second [$CI_{95\%}$: 12.13, 26.79] in the experimental condition. Thus, overall, subjects in both conditions crossed faster during the experimental phases than they did during the initial baseline measurement.

### Footfall errors

Posterior predictions for error rates for an average subject of each sex in all phases of the experiment are plotted in Fig 3A. These are computed based on the model's fixed effects, with an intercept equal to the population average. These parameters, along with their 95% posterior credible intervals are shown in Table 2. Overall, errors in the experimental group rise immediately following the experimental treatment, and remain elevated for the duration of the experiment. Error rates also

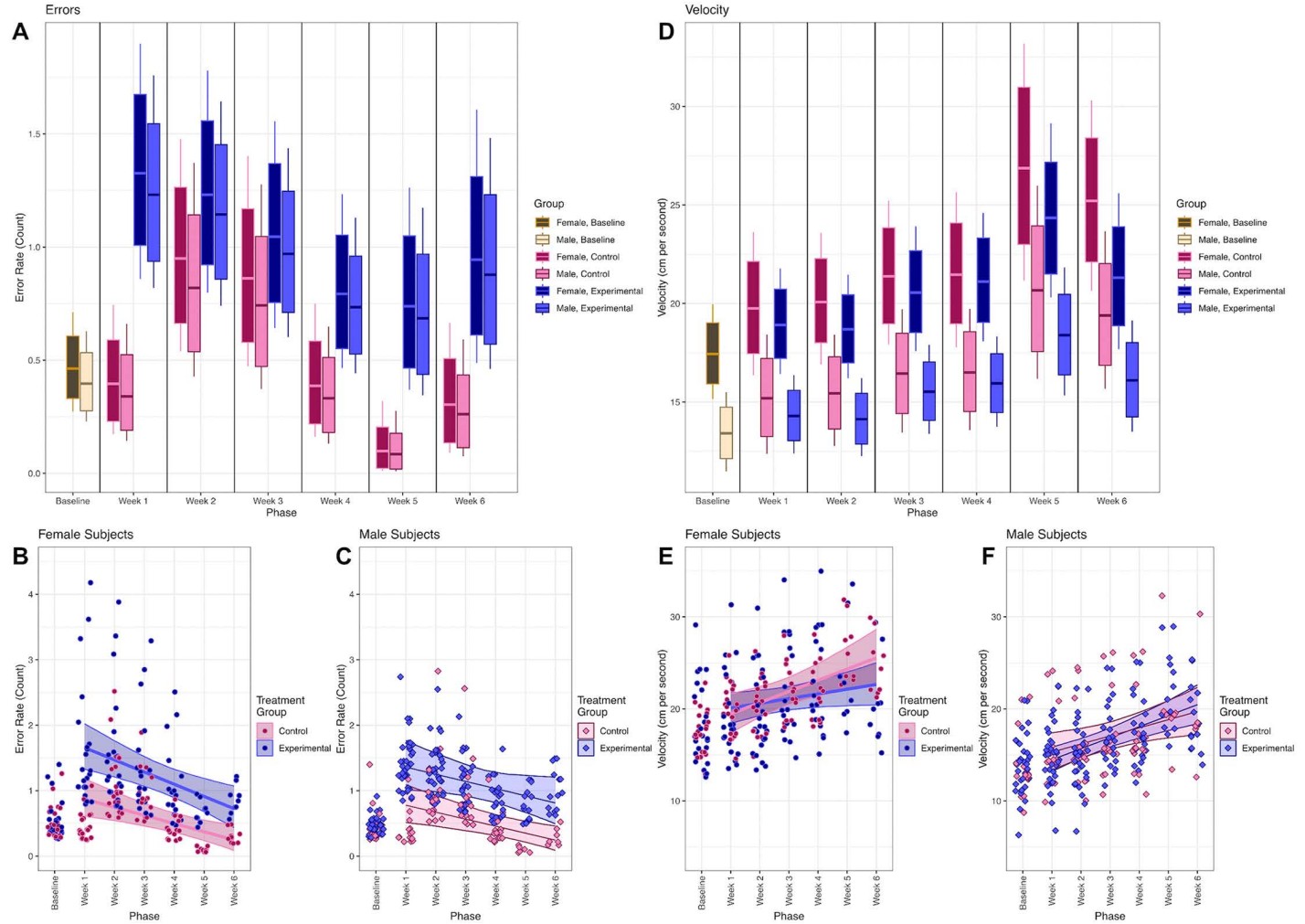

**Fig 3. Week-by-week estimates of performance. A.** Estimated error rate for an average subject during each week, split by condition (tan = base-line, pink = control, blue = experimental) and sex (darker colors = female, lighter colors = male), based on the population-level effects in Equations 1-4 (bars = 80% posterior credible interval, whiskers = 95%). **B.** Posterior mean estimates of individual error rates per session for each female subject, based on the full model in Equations 1-4. Each point reflects one subject during that session. Subjects are only plotted for weeks where their data are avail-able. Trendlines show the overall linear trend across subjects for the weeks after baseline (shaded region = 95% credible interval). **C.** As in panel C, but plotting male subjects. **D-F.** As in panels A-C, but plotting estimated velocities (cm/s).

rise in the control group, but only during weeks 2 and 3 of the study, falling back to or even below baseline levels in weeks 4–6. No reliable difference is observed due to sex.

The estimated error rates for each subject during every phase of the study in which they appeared are displayed in Fig 3B (females) and 3C (males). As such, estimates for all subjects appear at baseline and in weeks 1 and 2, but later weeks only include estimates for those cohorts still contributing data during those phases. Although the statistical model treated the six post-treatment phases as nominal variables for maximum flexibility, as noted above, we noted after fitting the data that subjects tended to show relatively gradual changes from week to week. Thus, we have also included a post-hoc estimate of the mean linear trend throughout the post-treatment phases for each group and each sex. These

**Table 2.** *Bayesian Model Parameters for Errors (log units).*

| Effect | Scale | Parameter | Mean | 95% CI |
|---|---|---|---|---|
| Pop. Intercept Mean | log(error rate) | $\eta$ | −0.875 | [-1.284, -0.502] |
| Pop. Intercept SD | log(error rate) | $\psi$ | 0.611 | [0.433, 0.817] |
| Sex | Female = −0.5, Male = 0.5 | $\beta_x$ | −0.160 | [-0.783, 0.464] |
| Sex × Experiment | Interaction | $\beta_{tx}$ | 0.092 | [-0.662, 0.878] |
| Week 1 | Dummy Code | $\beta_w[1]$ | −0.210 | [-1.007, 0.496] |
| **Week 2** | **~** | **$\beta_w[2]$** | ***0.707** | **[0.150, 1.262]** |
| **Week 3** | **~** | **$\beta_w[3]$** | ***0.597** | **[0.005, 1.188]** |
| Week 4 | ~ | $\beta_w[4]$ | −0.240 | [-1.068, 0.526] |
| **Week 5** | **~** | **$\beta_w[5]$** | ***−1.849** | **[-3.716, -0.372]** |
| Week 6 | ~ | $\beta_w[6]$ | −0.527 | [-1.609, 0.455] |
| **Week 1 × Experiment** | **Interaction** | **$\beta_t[1]$** | ***1.313** | **[0.597, 2.099]** |
| Week 2 × Experiment | ~ | $\beta_t[2]$ | 0.322 | [-0.220, 0.853] |
| Week 3 × Experiment | ~ | $\beta_t[3]$ | 0.263 | [-0.342, 0.887] |
| **Week 4 × Experiment** | **~** | **$\beta_t[4]$** | ***0.820** | **[0.014, 1.698]** |
| **Week 5 × Experiment** | **~** | **$\beta_t[5]$** | ***2.340** | **[0.826, 4.202]** |
| **Week 6 × Experiment** | **~** | **$\beta_t[6]$** | ***1.254** | **[0.191, 2.381]** |

Table 2 *Bayesian Model Parameters for Errors (log units)*. * Bolded/shaded rows are fixed effects whose posterior credible intervals exclude zero. Blue rows consistently make positive contributions, whereas orange rows consistently make negative ones. Rhat was approximately 1.00 for all the above parameters across 4000 posterior samples, with no ESS below 1400.

should be interpreted as a strictly descriptive overall summary of changes over time, reflecting the overall pattern of subjects becoming more experienced with the apparatus and/or recovering from the intervention.

In general, errors tended to fall over the course of the experiment: Post-hoc linear trends of change in errors per week consistently had negative slopes in most groups (control females: −0.123 [$CI_{95\%}$: −0.202, −0.053]; experimental females: −0.188 [$CI_{95\%}$: −0.292, −0.084]; control males: −0.106 [$CI_{95\%}$: −0.181, −0.038]), with the exception of experimental males (−0.112 [$CI_{95\%}$: −0.223, 0.003]). Overall, this reveals that although females overall had similar means to males, experimental females appeared slightly more variable, with larger outliers during the first few post-treatment weeks. Nevertheless, these results indicate that the variation in error rates was chiefly driven by a difference between the control and experimental conditions, without a consistent effect attributable to sex.

## Asymmetry analyses

An additional analysis evaluated whether error frequency differed between left and right paws (Equation 5; see the appendix for details). Subjects overall made consistently more errors with their left paws (74.0% [$CI_{95\%}$: 57.5%, 87.1%]). However, due to the low error rates overall, no other consistent differences could be detected, whether as a function of sex, treatment or phase. Independent cross-check verification of the original video recordings and the corresponding DLC-extracted data confirmed that the observed asymmetry was not attributable to differences in the software's detection of left- versus right-paw errors, nor to misclassifications.

## Velocity analyses

Posterior predictions of average velocities for males and females in all experimental phases are presented in Fig 3D, based on the fixed effects and a population-mean intercept. Model parameters, along with their 95% posterior credible

intervals, are provided in Table 3. Overall, subjects consistently complete the task more quickly following treatment than they did at baseline, and females traverse the apparatus more rapidly than males (including at baseline). No consistent treatment effect was observed, since weekly control group velocities closely matched those of the experimental group.

Estimated velocities for each subject across all phases of the study in which they appeared are shown in Fig 3E (female subjects) and 3F (male subjects), using the same post-hoc linear analysis described for Fig 3B & 3C. Once the post-treatment weeks are treated as ordered variables, a striking sex-by-treatment interaction emerges. Control females begin fast and also accelerate, their velocity increasing by 1.36 [$CI_{95\%}$: 0.48, 2.26] cm/second each week, whereas experimental females did not accelerate by as much (at 0.84 [$CI_{95\%}$: 0.21, 1.54] cm/second change each week). Males showed a similar pattern: Both control males (1.05 [$CI_{95\%}$: 0.38, 1.75] cm/second each week) and experimental males (0.64 [$CI_{95\%}$: 0.16, 1.16] cm/second each week) increased their velocity over time.

## TH-Asymmetry

Inter-rater agreement at the animal-side level supported averaging counts across raters. The single-measure absolute-agreement ICC was 0.841 ($CI_{95\%}$: 0.741, 0.904), and the average-measure ICC was 0.913 ($CI_{95\%}$: 0.851, 0.950). Bland-Altman analysis showed that the mean inter-rater difference was centered near zero, consistent with the absence of marked systematic bias, although dispersion was greater for the contralateral hemisphere and disagreements were present in a small subset of cases (see Supplementary Material 1). Final hemisphere-specific counts were therefore defined as the mean of the two raters.

Analysis of the $log(AI)$ showed a significant main effect of treatment, indicating lower ipsilateral-to-contralateral ratios in lesioned animals than in controls ($F(1, 21) = 43.78$, $p < 0.001$). Neither time ($F(2, 21) = 1.92$, $p = 0.171$) nor the treatment-by-time interaction ($F(2, 21) = 0.81$, $p = 0.460$) reached significance. These results indicate a lesion-related hemispheric asymmetry across cohorts without statistical evidence that its magnitude differed across t1, t2, and t3 (Fig 4).

**Table 3. *Bayesian Model Parameters for Velocity (log(cm/s)).***

| Effect | Scale | Parameter | Mean | 95% CI |
|---|---|---|---|---|
| Pop. Intercept Mean | log(velocity) | $\eta$ | 2.724 | [2.630, 2.824] |
| Pop. Intercept SD | log(velocity) | $\psi$ | 0.264 | [0.211, 0.327] |
| **Sex** | **Female = −0.5, Male = 0.5** | **$\beta x$** | ***−0.263** | **[-0.469, -0.065]** |
| Sex × Experiment | Interaction | $\beta_{tx}$ | −0.017 | [-0.276, 0.242] |
| Week 1 | Dummy Code | $\beta_w[1]$ | 0.117 | [-0.055, 0.287] |
| Week 2 | ~ | $\beta_w[2]$ | 0.137 | [-0.019, 0.292] |
| **Week 3** | **~** | **$\beta w[3]$** | ***0.198** | **[0.040, 0.356]** |
| **Week 4** | **~** | **$\beta w[4]$** | ***0.205** | **[0.051, 0.372]** |
| **Week 5** | **~** | **$\beta w[5]$** | ***0.423** | **[0.207, 0.635]** |
| **Week 6** | **~** | **$\beta w[6]$** | ***0.362** | **[0.175, 0.540]** |
| Week 1 × Experiment | Interaction | $\beta_t[1]$ | −0.041 | [-0.234, 0.156] |
| Week 2 × Experiment | ~ | $\beta_t[2]$ | −0.074 | [-0.254, 0.101] |
| Week 3 × Experiment | ~ | $\beta_t[3]$ | −0.040 | [-0.235, 0.148] |
| Week 4 × Experiment | ~ | $\beta_t[4]$ | −0.020 | [-0.220, 0.172] |
| Week 5 × Experiment | ~ | $\beta_t[5]$ | −0.095 | [-0.351, 0.169] |
| Week 6 × Experiment | ~ | $\beta_t[6]$ | −0.171 | [-0.407, 0.064] |

Table 3 **Bayesian Model Parameters for Velocity (log units).** Formatting conventions and convergence diagnostics are as in Table 2.

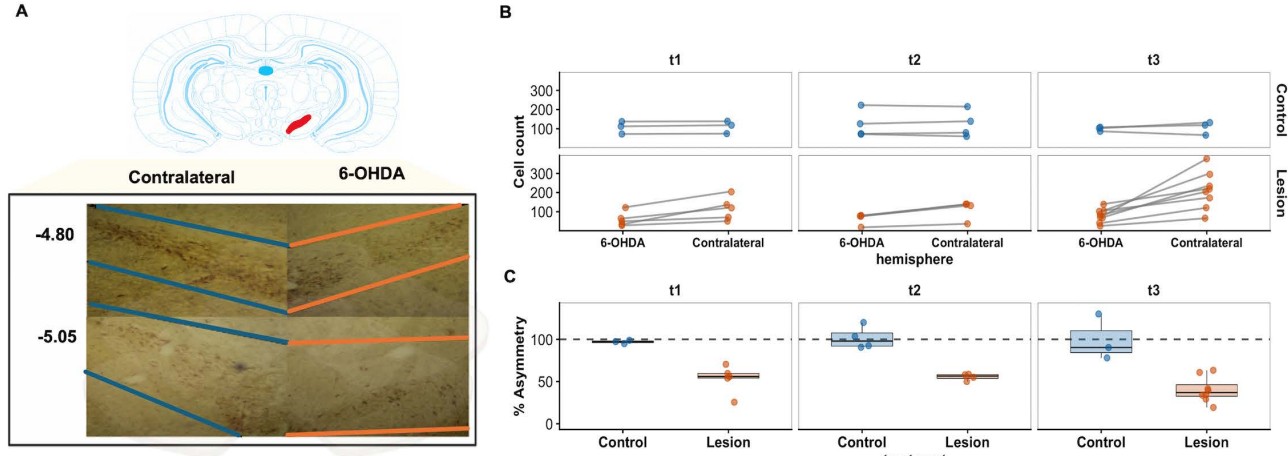

**Fig 4. Unilateral 6-OHDA lesion induces hemispheric asymmetry in nigral TH-positive cell counts across t1, t2 and t3. (A)** Reconstruction of the unilateral 6-OHDA infusion of the SNc. Representative coronal atlas schematics (top) and corresponding histological reconstruction (bottom) of the contralateral (blue) and 6-OHDA (orange) injected hemispheres. Colored lines delineate the extent of the substantia nigra region included in the analysis across serial sections at stereotaxic coordinates relative to bregma. **(B)** Final hemisphere-specific counts used to compute the Asymmetry Index (AI). Each point represents one animal, and paired lines connect right and left hemisphere counts from the same subject. Final counts were obtained by averaging tissue sections within each animal and hemisphere and then averaging across raters after inter-rater agreement had been evaluated (see Supplementary Material 1). Control animals showed limited hemispheric asymmetry, whereas lesioned animals showed lower counts in the injected hemisphere across cohorts. **(C)** Relative preservation percentage calculated from the Asymmetry Index across t1, t2, and t3 post-injection intervals. Relative preservation percentage was calculated as $AI \times 100$. Values close to 100% indicate similar counts across hemispheres, whereas lower values indicate reduced counts in the injected hemisphere. Boxplots show median and interquartile range.

Planned contrasts showed significant control-versus-lesion differences in $log(AI)$ at each time level (t1: estimate = 0.656, $p = 0.007$; t2: estimate = 0.603, $p = 0.009$; t3: estimate = 0.946, $p < 0.001$). Thus, the lesion effect was detectable at t1 and remained present at t2 and t3. Although the descriptive pattern suggested a larger effect at t3, this interpretation is limited by the absence of a significant treatment-by-time interaction.

Descriptive summaries of AI were consistent with the primary analysis. Control animals remained close to hemispheric symmetry across time, with mean relative preservation percentages of 97.1%, 101.5%, and 99.5% at t1, t2, and t3, respectively. In contrast, lesioned animals showed lower relative preservation of the injected hemisphere, with means of 53.1%, 55.4%, and 40.2% at t1, t2, and t3, respectively. Sensitivity analysis using the raw inter-hemispheric difference yielded the same overall pattern, with a significant main effect of treatment $F(1, 21) = 16.48$ (, $p < 0.001$), a trend to time effect $F(2, 21) = 3.22$ (, $p = 0.060$), and no treatment-by-time interaction ($F(2, 21) = 1.31$, $p = 0.292$).

## Discussion

The present study shows that partial unilateral dopaminergic depletion in the SNc induced by low-dose 6-OHDA produces subtle but persistent alterations in stepping performance and velocity dynamics in Wistar rats. This model generated a relatively mild behavioral phenotype characterized by sustained increases in footfall errors and more heterogeneous changes in velocity across time. While both lesioned and control males displayed a similar pattern of gradual increases in velocity across weeks, control females showed a more pronounced increase, whereas lesioned females exhibited a less marked and more variable pattern. Although the behavioral analysis was restricted to the two outcomes that were measured most robustly under the recording conditions used, these findings provide evidence of early changes in stepping accuracy and global locomotor progression in models of subthreshold dopamine depletion. Consequently, within

this framework, our results support the utility of partial SNc lesions for modeling early-stages of dopaminergic motor dysfunction.

Among the behavioral outcomes examined, footfall errors provided the clearest evidence of lesion-related impairment and revealed a nuanced pattern of gait alteration. Lesioned animals showed an early increase in stepping errors that remained elevated during the first two to three weeks after lesion, followed by partial improvement thereafter, whereas controls exhibited only a smaller and more transient increase before returning toward baseline levels. This pattern suggests that partial dopaminergic depletion affected rung grab accuracy in a sustained but temporally evolving manner. Although footfall error rates did not reveal a robust sex effect, lesioned females appeared more variable than lesioned males, which may reflect heterogeneity in behavioral adaptation; however, the possibility of differential sensitivity to dopaminergic depletion cannot be excluded [9]. Importantly, TH asymmetry was comparable across cohorts and sexes, suggesting that broad differences in lesion magnitude are less likely to fully account for this variability [20]. It is also worth noting that the analytical strategy prioritized comparability across sessions by selecting the least-interrupted trial, thereby reducing variability introduced by pauses not directly related to stepping execution. Although this approach may have reduced sensitivity to impairments expressed primarily as increased stopping or hesitation during ladder crossing, the temporal pattern of footfall errors still provided the most consistent behavioral indication that partial dopaminergic depletion disrupted stepping accuracy during the early post-lesion period, followed by a partial recovery.

This pattern of subtle but persistent impairment distinguishes the present model from more extensive 6-OHDA lesion protocols, in which severe nigrostriatal damage is typically accompanied by marked and progressive motor decline [21–23]. In our study, partial SNc lesions did not produce profound locomotor disability, but instead generated a milder behavioral profile in which stepping inaccuracies remained detectable across time and velocity changed in a more heterogeneous manner. These findings align with clinical observations in early PD, where gait alterations may precede overt bradykinesia [24]. This feature is particularly relevant from a translational perspective, because early motor dysfunction is difficult to be captured by models of widespread lesions in which deficits are more severe, evolve rapidly, and tend to be sustained over longer periods of time [25–27]. Rather, the present findings suggest that partial SNc lesions may be especially useful for examining how partial dopaminergic depletion dynamically alters motor performance during the interval in which compensatory processes and emerging dysfunction are likely to coexist, and before the broader deficits become more evident [28].

A further consideration is that the behavioral consequences of partial dopaminergic lesions may be shaped not only by neuronal loss itself, but also by early biological responses triggered by the lesion process. In 6-OHDA nigrostriatal models, partial or moderate dopaminergic degeneration has been associated with early microglial activation, reactive astrocytosis, apoptotic signaling, and measurable motor impairment, indicating that even relatively limited lesions unfold within a broader tissue response [29,30]. At the same time, this inflammatory component does not appear to be identical to that seen in more overtly inflammatory models such as LPS, as 6-OHDA has been associated with a lower degree of microglial activation and little evidence of systemic inflammatory engagement [30,31]. Within this framework, some of the temporal variability observed after partial dopaminergic depletion may reflect the interaction between lesion severity, early glial reactivity, and compensatory remodeling. In the present study, inflammatory markers were not assessed, and we therefore cannot determine whether neuroinflammatory processes contributed to the behavioral variability observed here. Nevertheless, this framework provides a plausible biological context for interpreting early heterogeneity in gait adaptation after partial lesions and represents an important direction for future work.

The subtle and temporally dynamic nature of the impairment observed here is consistent with the view that early dopaminergic dysfunction is not expressed as uniform deterioration, but rather as an interplay between deficit, adaptation, and partial behavioral recovery [32,33]. In this context, the overall increase in footfall errors alongside a more variable pattern in velocity may indicate that partial lesions affect stepping precision more readily than global locomotor progression. Likewise, a reduction in velocity was descriptively more apparent in lesioned females relative to their control counterparts,

whereas footfall errors increased in both sexes. Because velocity was intended to capture global progression, our results should be interpreted as an index of overall movement across the ladder rather than as an exhaustive detailed stride-by-stride kinematics. Even so, the dissociation between relatively persistent stepping inaccuracies and less uniform changes in velocity is conceptually compatible with reports from early-stage human PD cohorts [34–36] in which subtle changes in gait consistency may precede overt slowing.

Although some results suggested different degrees of variability across males and females, these patterns were not uniformly robust across outcomes, and some estimates showed substantial uncertainty. Accordingly, the present findings might suggest possible sex-related heterogeneity rather than demonstrating sex-dependent motor phenotypes. This caution is especially important because the experimental design was not targeted to isolate biological sources of sex-related variance. The estrous cycle was not controlled, and the sample distribution across sex, treatment, and terminal cohorts may also have contributed to uncertainty in some comparisons. Under these conditions, the greater variability observed lesioned females suggest that early behavioral responses to partial dopaminergic depletion may be more heterogeneous than group averages alone indicate. Accordingly, within the broader PD literature, sex-related differences in onset, progression, and compensatory responses remain biologically plausible [37–40], and hormonal influences such as estrogen signaling may represent one of several candidate mechanisms that may contribute to these variability. Further research is needed to establish whether partial dopaminergic lesion models give rise to differential behavioral trajectories associated with sex, and to clarify the role of hormonal influences during early stages of dopaminergic dysfunction and across Parkinson's disease progression.

The temporal profile of behavioral change also deserves consideration in light of the study design. Because animals were distributed across terminal cohorts defined by histological endpoints, the six-week post-lesion pattern reflects group-level trajectories derived from partially overlapping cohorts rather than continuous within-subject follow-up across the full interval for every animal. This design was necessary to relate behavioral evolution to TH-defined lesion status at multiple post-lesion stages, but it also means that temporal patterns should be interpreted as aggregated longitudinal profiles rather than as complete individual trajectories. Even so, the sustained elevation of footfall errors in lesioned animals, compared with the more transient changes observed in controls, argues against a purely procedural explanation. However, the persistence and relative stability of errors across weeks indicate that dopaminergic depletion contributed to the observed behavioral profile beyond other procedural factors. Within this context, characterizing gait alterations during early dopaminergic depletion requires moving beyond conventional approaches to motor impairment, in which the magnitude of errors is often treated as the main indicator of motor disability. Instead, these early models could be more accurately characterized by considering the temporal profile of errors, including their persistence and patterns of resolution, together with changes in velocity over time. These dynamic features may help refine the characterization of early trajectories of dopaminergic dysfunction and strengthen the translational relevance of preclinical models [41,42].

The use of DeepLabCut was an important methodological strength of the study. Under the present recording conditions, markerless tracking enabled standardized quantification of velocity and footfall events using accessible video hardware, thereby improving the consistency of behavioral scoring without requiring specialized motion-capture systems [16,17,43]. Nevertheless, this configuration imposed some technical limits. Recordings were obtained from a single right-sided lateral view at 30 fps, which constrained the extraction of finer kinematic variables and may have reduced sensitivity for detecting some subtle left paw errors after a right-hemisphere lesion. For this reason, the present analysis focused on two locomotor outcomes that were extracted and interpreted reliably under these conditions rather than on a more exhaustive description of gait. Thus, the methodological contribution of the study lies in showing that locomotor progression across the apparatus and stepping accuracy can be quantified in a relatively sensitive and scalable manner in a partial-lesion model.

Taken together, the study was designed to capture longitudinal changes in selected locomotor outcomes under recording conditions optimized for consistency and feasibility. Low-dose unilateral 6-OHDA lesions in the SNc produced subtle

yet sustained impairments in stepping accuracy together with more variable changes in velocity, a pattern consistent with an early stage of dopaminergic motor dysfunction rather than severe motor disability that fully characterized clinical Parkinson's. These results support the use of partial SNc lesions as a useful framework for investigating early motor alterations and their temporal evolution, while also showing that markerless DLC–based tracking can improve the resolution of gross locomotor outcomes in such models under accessible experimental conditions. Finally, the present findings reinforce the importance of systematically including both sexes in preclinical PD research. Even when sex effects are not uniformly robust, their possible contribution to early behavioral heterogeneity is itself biologically informative and warrants future studies specifically designed to evaluate whether motor alteration and recovery follow different trajectories across sexes, as well as which inflammatory, glial, hormonal, and other neural or non-neural mechanisms may shape those trajectories after partial dopaminergic depletion.

## Supporting information

**S1 File. Assessment of inter-rater agreement.**
(DOCX)

**S2 File. Bayesian Modeling.**
(DOCX)

## Acknowledgments

We thank Yaneth Camargo and Yaneth Gómez for their support in animal care and for maintaining laboratory operations during the pandemic closure. We are particularly grateful to Dr. Keydy Vasquez and Dr. Camilo Salamanca, veterinarians at the Animal Core Facility of the University of Los Andes, for their generous support, expert guidance, and training. We also thank the undergraduate and graduate students of the Neuroscience and Behavior Laboratory for their assistance with manual video editing and cell counting, as well as for their thoughtful feedback throughout the study.

## Author contributions

**Conceptualization:** Diego Lievano Parra, Greg Jensen, Valeria Gonzalez Diaz, Fernando Cardenas Parra.

**Data curation:** Diego Lievano Parra, Greg Jensen.

**Formal analysis:** Diego Lievano Parra, Juan David Garavito Coronado, Greg Jensen.

**Funding acquisition:** Fernando Cardenas Parra.

**Investigation:** Diego Lievano Parra, Fernando Cardenas Parra.

**Methodology:** Diego Lievano Parra, Fernando Cardenas Parra.

**Project administration:** Fernando Cardenas Parra.

**Resources:** Fernando Cardenas Parra.

**Software:** Juan David Garavito Coronado.

**Supervision:** Fernando Cardenas Parra.

**Validation:** Diego Lievano Parra, Juan David Garavito Coronado, Greg Jensen, Valeria Gonzalez Diaz.

**Visualization:** Diego Lievano Parra, Juan David Garavito Coronado, Greg Jensen.

**Writing – original draft:** Diego Lievano Parra, Juan David Garavito Coronado, Greg Jensen, Valeria Gonzalez Diaz.

**Writing – review & editing:** Diego Lievano Parra, Greg Jensen, Valeria Gonzalez Diaz.

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
