## [Decision Letter · Decision Letter 0]

18 Feb 2026

PONE-D-25-59811Sex-specific trajectories of gait adaptation following partial dopaminergic lesions in a rat model of parkinson’s diseasePLOS One

Dear Dr. Lievano Parra,

Thank you for submitting your manuscript to PLOS ONE. After careful consideration, we feel that it has merit but does not fully meet PLOS ONE’s publication criteria as it currently stands. Therefore, we invite you to submit a revised version of the manuscript that addresses the points raised during the review process.

Thank-you for submitting your work. The reviewers have provided extensive comments on your manuscript. We hope that you can revise and respond to each comment in detail. We look forward to your revised manuscript.

We look forward to receiving your revised manuscript.

Kind regards,

Nafisa M. Jadavji, PhD, MSc, BSc

Academic Editor

PLOS One

**Journal Requirements:**

1. When submitting your revision, we need you to address these additional requirements. Please ensure that your manuscript meets PLOS ONE's style requirements, including those for file naming. The PLOS ONE style templates can be found at https://journals.plos.org/plosone/s/file?id=wjVg/PLOSOne_formatting_sample_main_body.pdf and https://journals.plos.org/plosone/s/file?id=ba62/PLOSOne_formatting_sample_title_authors_affiliations.pdf 2. Please note that PLOS One has specific guidelines on code sharing for submissions in which author-generated code underpins the findings in the manuscript. In these cases, we expect all author-generated code to be made available without restrictions upon publication of the work. Please review our guidelines at https://journals.plos.org/plosone/s/materials-and-software-sharing#loc-sharing-code and ensure that your code is shared in a way that follows best practice and facilitates reproducibility and reuse. 3. In the online submission form, you indicated that your data will be submitted to a repository upon acceptance.  We strongly recommend all authors deposit their data before acceptance, as the process can be lengthy and hold up publication timelines. Please note that, though access restrictions are acceptable now, your entire minimal  dataset will need to be made freely accessible if your manuscript is accepted for publication. This policy applies to all data except where public deposition would breach compliance with the protocol approved by your research ethics board. If you are unable to adhere to our open data policy, please kindly revise your statement to explain your reasoning and we will seek the editor's input on an exemption. 4. When completing the data availability statement of the submission form, you indicated that you will make your data available on acceptance. We strongly recommend all authors decide on a data sharing plan before acceptance, as the process can be lengthy and hold up publication timelines. Please note that, though access restrictions are acceptable now, your entire data will need to be made freely accessible if your manuscript is accepted for publication. This policy applies to all data except where public deposition would breach compliance with the protocol approved by your research ethics board. If you are unable to adhere to our open data policy, please kindly revise your statement to explain your reasoning and we will seek the editor's input on an exemption. Please be assured that, once you have provided your new statement, the assessment of your exemption will not hold up the peer review process. 5. If the reviewer comments include a recommendation to cite specific previously published works, please review and evaluate these publications to determine whether they are relevant and should be cited. There is no requirement to cite these works unless the editor has indicated otherwise.

Reviewers' comments:

Reviewer's Responses to Questions

**Comments to the Author**

1. Is the manuscript technically sound, and do the data support the conclusions?

Reviewer #1: Yes

Reviewer #2: Yes

2. Has the statistical analysis been performed appropriately and rigorously? 

Reviewer #1: Yes

Reviewer #2: Yes

3. Have the authors made all data underlying the findings in their manuscript fully available?

Reviewer #1: Yes

Reviewer #2: Yes

4. Is the manuscript presented in an intelligible fashion and written in standard English?

Reviewer #1: Yes

Reviewer #2: Yes

5. Review Comments to the Author

**Reviewer #1:**Dear authors,

I deeply appreciate the opportunity to review this interesting and methodologically rich manuscript. The study addresses an important topic (the emergence of sex-specific gait adaptations following partial dopaminergic lesions) and combines DeepLabCut-based kinematics with histological validation and Bayesian modeling. This is a promising direction, especially for modeling early-stage Parkinsonian motor changes.

Below, I would like to offer suggestions that can strengthen the clarity, methodological transparency, and interpretability of your work.

Main Comments

1. Experimental design and cohort structure

The rationale for using three independent post-lesion cohorts (t1, t2, t3) rather than repeated measures in the same animals could be articulated more explicitly. This decision has consequences for power and for the interpretation of trajectories.

Please also report the exact sample distribution by sex × treatment × time, ideally in a table, to provide a better understanding.

2. Behavioral analysis would benefit from additional validation and explanation

Authors evaluate two behavioral outcomes (footfall errors and velocity). Given the capacity of DLC to extract richer gait metrics, the rationale for this restriction should be more clearly presented in the Methods.

Additionally, velocity remains in pixel-based units; this limits interpretation across animals and time points. Clarifying the absence of spatial calibration and discussing its implications would improve the manuscript.

3. Statistical modeling is sophisticated but needs clearer reporting

The Bayesian hierarchical model is a strength of the study. I have some concerns:

• equations are difficult for general readers to follow

• model diagnostics (Rhat, ESS, trace plots, posterior predictive checks) could be reported

• post-lesion weeks are modeled as categorical factors, but interpreted temporally—this mismatch needs discussion

As a suggestion, schematic explanation of the model’s rationale could help make this section more accessible.

4. Interpretation of sex differences should be more cautious

Some sex-related effects show overlapping 95% credible intervals or inconsistent patterns across variables.

Before proposing mechanistic explanations (e.g., estrogen effects), I recommend clarifying:

• whether the design adequately controls for estrous cycle variability

• whether sample imbalance between sexes may influence uncertainty

• which effects are robust vs. trending

5. DeepLabCut training and validation details

Concerning DeepLabCut, only 303 frames were used to train a ResNet-152 model, which is relatively small for a deep network.

In order to provide a better understanding to the reader, some points could be clarified:

• how training frames were selected

• whether frames included all gait phases

• how model performance was evaluated

• whether labeling consistency was assessed

6. Figures

The figures are informative. I just would suggest that red–green contrasts be avoided to improve readability for color-blind readers.

7. TH quantification methods require more detail

As TH asymmetry anchors the biological interpretation, please consider to include:

• how SNc ROIs were defined

• how optical density was normalized

• whether quantification followed stereological guidelines

• how hemispheric asymmetry was calculated

These additions will reinforce the histological–behavioral link.

8. Discussion

Consider discussing the contribution of neuroinflammatory processes to partial dopaminergic lesions.

The Discussion would benefit from acknowledging that partial or subthreshold dopaminergic lesions commonly induce early neuroinflammatory responses in the model adopted, which can influence motor adaptation, synaptic remodeling, and even sex-dependent vulnerability.

Including this perspective would enrich the interpretation of your findings, especially regarding variability in gait adaptation. Recent work has highlighted the functional relevance of early inflammatory modulation in partial lesion models (https://doi.org/10.3390/neuroglia6030036), which should be included. A brief paragraph contextualizing this would strengthen the mechanistic framework of the Discussion.

Minor Comments

• Week/time labels (t1, t2, t3) should be standardized across all figures and tables.

• Please check for minor typographical issues (e.g., “Gonazalez”).

**Reviewer #2:** The aim of this study was to capture early motor deficits in a low-dose 6-OHDA rat model of Parkinsons disease. The authors combined longitudinal behavioral assessment on a horizontal ladder and a Bayesian modelling. The approach is innovative and results are also promising. However, same methodological and interpretations concerns should be explained before this manuscript can be considered for publication.

Major revision

1. For each session, the authors selected the trial with the fewest stops, or the first uninterrupted crossing. This procedure could introduce a systematic selection bias considering that stopping behaviour may represent indication of motor impairment in this lesion model. This issue should be explained or addressed in the Discussion.

2. The manuscript presents two different definitions of locomotor velocity. In the behavioral task description, velocity is “calculated from the time the four paws first contacted the rungs until the head reached the end of the ladder”, whereas in the DLC-based analysis and statistical modelling, velocity is “calculated using the displacement of the tail base across frames and reported in pixels per second (px/s)”. It should be clarified which metric were used to the reposted results.

3. In the Discussion section, the authors emphasis on sex-specific adaptive strategies and differential sensitivity to dopaminergic depletion. However, results presented do not reveal statistically supported effects of sex for footfall errors. These statements should be reformulated more cautiously in this section.

4. The authors state that this study aims to quantify stepping errors and overall locomotor velocity rather than to perform a detailed kinematic analysis of gait. In this case it is reasonable the use of a single lateral camera at 30 fps for their behavioral analysis. However, it is noteworthy that, despite the lesion being performed in the right hemisphere, locomotor recordings were obtained from the right side of the animal. This camera placement could potentially reduce sensitivity and accuracy for detecting left paw footfall errors, which should be clarified by the authors.

Minor

Figure 1B. The images shown should be better explained in the figure caption. The labels C and L in the microphotographs are not defined.

6. PLOS authors have the option to publish the peer review history of their article (what does this mean?). If published, this will include your full peer review and any attached files.

Reviewer #1: No

Reviewer #2: No

---

## [Author Response · Author response to Decision Letter 1]

6 Apr 2026

Reviewer 1. Comment 1. Experimental design and cohort structure. The rationale for using three independent post-lesion cohorts (t1, t2, t3) rather than repeated measures in the same animals could be articulated more explicitly. This decision has consequences for power and for the interpretation of trajectories. Please also report the exact sample distribution by sex × treatment × time, ideally in a table, to provide a better understanding.

Response: We thank you for pointing this out. We have revised the Methods to clarify the rationale for using three independent post-lesion cohorts. Animals were assigned to terminal endpoints at 2 (t1), 4(t2), or 6(t3) weeks post-lesion because quantification of TH immunoreactivity required tissue collection at each interval, precluding complete within-subject follow-up across the full 6-week period. Thus, behavioral measures were obtained longitudinally within each cohort until its designated endpoint, whereas the overall post-lesion time course reflects group-level temporal patterns derived from partially overlapping cohorts. We now make this interpretive constraint explicit in the revised manuscript. In addition, to improve transparency, we have included a table reporting the exact sample distribution by sex × treatment × cohort/time point, as requested (Methods, lines 127-142; Discussion, lines 618-6525).

Reviewer 1. Comment 2. Behavioral analysis would benefit from additional validation and explanation. Authors evaluate two behavioral outcomes (footfall errors and velocity). Given the capacity of DLC to extract richer gait metrics, the rationale for this restriction should be more clearly presented in the Methods. Additionally, velocity remains in pixel-based units; this limits interpretation across animals and time points. Clarifying the absence of spatial calibration and discussing its implications would improve the manuscript.

Response: We have revised the Methods to clarify why the DeepLabCut analysis focused on velocity and footfall errors. Because videos were acquired using a single lateral camera, the analysis was restricted to measures that could be extracted consistently and interpreted reliably under these recording conditions. Accordingly, we focused on velocity and footfall errors, which captured locomotor progression and stepping accuracy, respectively. Regarding velocity units, in the revised manuscript, velocity has now been converted from pixel-based values to cm/s for all videos using a spatial calibration based on the known 4-cm distance between adjacent rungs of the apparatus. The Methods have been updated to describe this calibration procedure, and the Results now report velocity in cm/s accordingly (Methods, lines 181-192; 295-307). We also clarify the interpretive scope of this measure in the Discussion (lines 526-533; 642-650).

Reviewer 1. Comment 3. Statistical modeling is sophisticated but needs clearer reporting. The Bayesian hierarchical model is a strength of the study. I have some concerns:

● Equations are difficult for general readers to follow.

● Model diagnostics (Rhat, ESS, trace plots, posterior predictive checks) could be reported.

● Post-lesion weeks are modeled as categorical factors, but interpreted temporally—this mismatch needs discussion.

Response: Balancing the complexity of non-linear hierarchical modeling with legibility to a general audience is always tricky, and we appreciate the recognition that our modeling efforts are seen as a strength. We have made several adjustments to hopefully facilitate reader comprehension (Methods, lines 309-334). Primarily, we have moved the bulk of the model specification (priors, link functions, Stan parameters, etc.) to an appendix at the end of the text, where we have provided a bit more description of the particulars (Appendix, lines 667-709). Our hope is that this will be less disruptive to the flow of the paper for general readers.

As we now note in Tables 2 and 3, Rhat was 1.00 and ESS was at least 1000 for all parameter estimates (given 4000 posterior samples), providing plenty of confidence that we could characterize the 95% credible interval (Results, lines 382-3, 464-5). Given the number of parameters in the model, we elected not to include trace plots; we are confident our high ESS sufficiently conveys that we had adequate mixing without needing to include a figure with over two dozen panels that general readers would not know how to interpret.

Additionally, we have effectively provided a kind of posterior prediction check, at least implicitly, in the contrast between Figure 2 (which reports data for all subjects) and Figure 3 (which reports posterior mean estimates for all subjects). Given the discrete nature of Poisson data, especially when rate parameters are low, we did not feel that taking the final step of simulating data with residual error would be helpful to the reader.

The reviewers point out, correctly, that there is an evident mismatch between the model (which treats time nominally) and our linear trendlines. We have clarified in the manuscript that these trendlines are only included as a descriptive summary, reflecting gross patterns in the data post-manipulation (Results, lines 428-434). Our feeling is that most readers would trace these functions by eye anyway. Although we considered nonlinear alternatives (such as LOESS smoothers or splines), the reduction in the sample size over time makes extrapolation much riskier. Our hope is that a linear trendline, presented with the added caveats that it is included purely for descriptive purposes, will be more helpful to the reader that merely plotting the points alone.

Reviewer 1. Comment 4. Interpretation of sex differences should be more cautious. Some sex-related effects show overlapping 95% credible intervals or inconsistent patterns across variables.

Before proposing mechanistic explanations (e.g., estrogen effects), I recommend clarifying:

● Whether the design adequately controls for estrous cycle variability.

● Whether sample imbalance between sexes may influence uncertainty.

● Which effects are robust vs. trending.

Response: We agree that the interpretation of sex-related effects requires greater caution. The revised Discussion distinguish more clearly between patterns that were consistently supported by the data and those that should be interpreted more tentatively. Specifically, we now state that footfall errors did not provide strong statistical support for a sex effect, and that the differences observed were expressed mainly as variability patterns rather than as clear sex-dependent motor phenotypes (Discussion, lines 539-546; 593-599). We also clarify that the study was not designed to isolate biological sources of sex-related variance, as estrous cycle was not controlled and the sample distribution across sex, treatment, and terminal cohorts may have contributed to uncertainty in some comparisons (Discussion, lines 600-610). Accordingly, mechanistic interpretations involving estrogen-related protection or sex-dependent compensatory processes are now framed as hypotheses for future targeted studies rather than as conclusions supported directly by the present dataset. These revisions have been incorporated into the Discussion (lines 610-617).

Reviewer 1. Comment 5. DeepLabCut training and validation details concerning DeepLabCut, only 303 frames were used to train a ResNet-152 model, which is relatively small for a deep network. In order to provide a better understanding to the reader, some points could be clarified:

● How training frames were selected

● Whether frames included all gait phases

● How model performance was evaluated

● Whether labeling consistency was assessed

Response: We have expanded the Methods to clarify the DeepLabCut training and validation procedure. Training frames were manually selected from multiple videos to capture a representative range of postures, locomotor states, and gait-related conditions observed in the dataset, including forward progression, stationary postures, slips, and hopping-like movements. In addition, frame selection was intended to sample multiple limb configurations across the locomotor cycle, including paw contact, limb lift, and forward swing, although frames were not formally categorized into predefined gait phase classes. This clarification has now been added to the Methods (lines 199-204). We also now clarify that, although the network backbone was based on ResNet-152, DeepLabCut relies on transfer learning from models pre-trained on large image datasets, which substantially reduces the number of manually annotated frames required for accurate pose estimation. In this study, 303 annotated frames were used to provide broad coverage of the observed behavioral variability (Methods, lines 271-274).

Model performance was evaluated using the mean absolute error (MAE) between predicted and manually annotated landmark positions in both training and test sets. The trained network achieved a training MAE of 3.80 px and a test MAE of 8.41 px, which decreased to 5.42 px after applying a likelihood threshold of ≥ 0.75, indicating accurate landmark prediction suitable for downstream gait analysis. We have added these values and a brief explanation of the likelihood threshold to the revised Methods. Finally, we clarified that landmark annotations were performed using fixed labeling criteria and reviewed for consistency across frames representing different postures and locomotor conditions. Landmarks were assigned only when the corresponding body part was clearly visible and within the frame. These clarifications have now been incorporated into the revised manuscript (Methods, lines 277-285)

Reviewer 1. Comment 6. Figures. The figures are informative. I just would suggest that red–green contrasts be avoided to improve readability for color-blind readers.

Response: We have swapped green out for tan hues and performed level checks to verify that the color scheme is interpretable even viewed as strict grayscale.

Reviewer 1. Comment 7. TH quantification methods require more detail as TH asymmetry anchors the biological interpretation, please consider to include:

● How SNc ROIs were defined.

● How optical density was normalized.

● Whether quantification followed stereological guidelines.

● How hemispheric asymmetry was calculated.

Response: We appreciate this comment and agree that the TH quantification procedure needed to be described in greater detail, particularly because the asymmetry measure is central to the biological interpretation of the lesion. We have revised the Methods section accordingly. In the revised version, we specify how SNc sections were identified using the Paxinos and Watson rat brain atlas and how ROIs were defined in ImageJ based on atlas-guided anatomical boundaries and consistent cytoarchitectural landmarks across hemispheres and animals. We also clarify that TH quantification was based on direct manual counts of TH-positive somatic profiles within these ROIs, performed independently by two evaluators blinded to treatment condition using the Cell Counter tool in ImageJ. Fibers and diffuse staining were excluded from the analysis. We also added information on inter-rater agreement (Methods, lines 216-224; 236-256 Supplementary material 1).

We further add that optical density measurements were not used in this study, as the analysis was based on direct cell counts. Likewise, stereological methods were not applied because our goal was not to estimate absolute neuronal number, but rather to quantify relative hemispheric depletion within each animal (Methods, lines 253-256).

Finally, we now define hemispheric asymmetry explicitly using an Asymmetry Index (AI = I/C), where the injected hemisphere is expressed relative to the contralateral side, and we specify that the primary inferential outcome was the log-transformed AI analyzed as a function of treatment, time, and their interaction (Methods, lines 249-268).

Reviewer 1. Comment 8. Discussion. Consider discussing the contribution of neuroinflammatory processes to partial dopaminergic lesions. The Discussion would benefit from acknowledging that partial or subthreshold dopaminergic lesions commonly induce early neuroinflammatory responses in the model adopted, which can influence motor adaptation, synaptic remodeling, and even sex-dependent vulnerability. Including this perspective would enrich the interpretation of your findings, especially regarding variability in gait adaptation. Recent work has highlighted the functional relevance of early inflammatory modulation in partial lesion models (https://doi.org/10.3390/neuroglia6030036), which should be included. A brief paragraph contextualizing this would strengthen the mechanistic framework of the Discussion.

Response: We agree that the contribution of neuroinflammatory processes merits acknowledgment in the context of partial dopaminergic lesion models. Although the present study did not directly assess inflammatory markers, we acknowledge that early glial and inflammatory responses have been described in toxin-based models of nigrostriatal degeneration and may shape behavioral adaptation during the initial phases of lesion development. We have revised the Discussion accordingly to incorporate this mechanistic perspective in a more explicit way. We have also cited the suggested reference, together with additional relevant literature, to place our findings within the broader context of early inflammatory modulation in 6-OHDA models (Discussion, lines 569-585).

Reviewer 1. Minor Comments

• Week/time labels (t1, t2, t3) should be standardized across all figures and tables.

• Please check for minor typographical issues (e.g., “Gonazalez”).

Response: We apologize for the mistakes that have now been corrected

Reviewer 2. Comment 1. For each session, the authors selected the trial with the fewest stops, or the first uninterrupted crossing. This procedure could introduce a systematic selection bias considering that stopping behaviour may represent indication of motor impairment in this lesion model. This issue should be explained or addressed in the Discussion.

Response: We thank you for raising this point. The purpose of selecting the trial with the fewest stops, or the first uninterrupted crossing when available, was to evaluate stepping accuracy and velocity under conditions of continuous locomotion. Because pauses during ladder crossing can reflect a mixture of exploratory behavior, hesitation, and task disengagement in addition to motor difficulty, trials with fewer interruptions were used to obtain a more standardized estimate of rung negotiation performance. We agree, however, that stopping behavior may also capture a meaningful component of motor impairment in this lesion model. Accordingly, we now clarify in the Methods the rationale for this trial-selection strategy and acknowledge in the Discussion that this procedure may underrepresent deficits expressed primarily as increased stopping during traversal (Methods, lines 177-184; Discussion, lines 546-553).

Reviewer 2. Comment 2. The manuscript presents two different definitions of locomotor velocity. In the behavioral task description, velocity is “calculated from the time the four paws first contacted the rungs until the head reached the end of the ladder”, whereas in the DLC-based analysis and statistical modelling, velocity is “calculated using the displacement of the tail base across frames and reported in pixels per second (px/s)”. It should be clarified which metric were used to the reposted results.

Response: We thank you for identifying this point. We have already clarified that the results reported in the manuscript are based on a single velocity measure. In all analyses, trial duration was defined from the onset of forward movement on the ladder to the frame in which the animal reached the end of the apparatus, and distance was estimated from the horizontal displacement of the ear-base landmark across that interval. Velocity was calculated as displacement over time and is now reported consistently in cm/s in the revised manuscript. We have therefore revised the text to eliminate the inconsistent wording in the original submission and to provide a unified definition of velocity across t

---

## [Decision Letter · Decision Letter 1]

27 Apr 2026

Time-dependent changes in stepping performance and velocity following partial dopaminergic lesions in the SNc of male and female rats

PONE-D-25-59811R1

Dear Dr. Lievano Parra,

We’re pleased to inform you that your manuscript has been judged scientifically suitable for publication and will be formally accepted for publication once it meets all outstanding technical requirements.

Kind regards,

Nafisa M. Jadavji, PhD, MSc, BSc

Academic Editor

PLOS One

Additional Editor Comments (optional):

Reviewers' comments:

Reviewer's Responses to Questions

**Comments to the Author**

1. If the authors have adequately addressed your comments raised in a previous round of review and you feel that this manuscript is now acceptable for publication, you may indicate that here to bypass the “Comments to the Author” section, enter your conflict of interest statement in the “Confidential to Editor” section, and submit your "Accept" recommendation.

Reviewer #1: All comments have been addressed

Reviewer #2: All comments have been addressed

2. Is the manuscript technically sound, and do the data support the conclusions?

Reviewer #1: Yes

Reviewer #2: Yes

3. Has the statistical analysis been performed appropriately and rigorously? 

Reviewer #1: Yes

Reviewer #2: Yes

4. Have the authors made all data underlying the findings in their manuscript fully available?

Reviewer #1: Yes

Reviewer #2: Yes

5. Is the manuscript presented in an intelligible fashion and written in standard English?

Reviewer #1: Yes

Reviewer #2: Yes

6. Review Comments to the Author

Reviewer #1: (No Response)

Reviewer #2: The authors have made substantial and constructive revisions that significantly improve the clarity, transparency, and overall comprehensibility of the study. All concerns raised by this reviewer have been adequately addressed.

The authors have clarified the definition of velocity, discussed the limitations associated with camera placement and the detection of contralateral errors, and revised the interpretation of sex-related effects to a more cautious conclusion. Finally, the limitations of the experimental design have been appropriately discussed in the manuscript.

In conclusion, I believe the manuscript is now suitable for publication in its current form.

7. PLOS authors have the option to publish the peer review history of their article (what does this mean?). If published, this will include your full peer review and any attached files.

Reviewer #1: No

Reviewer #2: No

---

## [Editor Report · Acceptance letter]

PONE-D-25-59811R1

PLOS One

Dear Dr. Lievano Parra,

I'm pleased to inform you that your manuscript has been deemed suitable for publication in PLOS One. Congratulations! Your manuscript is now being handed over to our production team.

Kind regards,

on behalf of

Dr. Nafisa M. Jadavji

Academic Editor

PLOS One